# Human Appropriation of Net Primary Production Related to Livestock Provisioning Ecosystem Services in Southern Patagonia

Pablo L. Peri [1,2,*], Yamina M. Rosas [3] and Guillermo Martínez Pastur [3]

1 Instituto Nacional de Tecnología Agropecuaria (INTA), Río Gallegos 9400, Santa Cruz, Argentina
2 Department of Natural Resources, Universidad Nacional de la Patagonia Austral (UNPA)—CONICET, Río Gallegos 9400, Santa Cruz, Argentina
3 Laboratorio de Recursos Agroforestales, Centro Austral de Investigaciones Científicas (CADIC CONICET), Ushuaia 9410, Tierra del Fuego, Argentina; yamicarosas@gmail.com (Y.M.R.); cadicforestal@gmail.com (G.M.P.)
* Correspondence: peri.pablo@inta.gob.ar

**Abstract:** Human appropriation of net primary productivity (HANPP) integrates ecological and socioeconomic perspectives on land use by quantifying the amount of net primary production (NPP) appropriated by society through biomass harvest from the ecosystem. The main objective of this study was to determine the spatial patterns of HANPP related to lamb and wool production from sheep farms across the province of Santa Cruz. The HANPP was obtained by dividing the sum of the biomass used in livestock products (lamb and wool) by the NPP. In addition, we examined the spatial relationship between HANPP and potential plant biodiversity and net carbon balance at the farm level under livestock land use across our study region. At the regional level, livestock production accounted for an average of 11.35% of appropriated NPP, and HANPP ranged from 0.75 to 50%. The map of HANPP across Santa Cruz showed low values in the vegetation transition (ecotone) between *Nothofagus antarctica* forests and grasslands in the west, in the south, and in wetlands where the most productive rangelands dominate. High values were observed in the northwest and central areas of the province. There were differences in HANPP across vegetation types with mean values that varied from 3.93% in grasslands on the Humid Magellanic Steppe to 12.33% in the Central Plateau. Simple linear regression analysis for HANPP evaluated in Southern Patagonia showed a negative linear relationship ($p < 0.05$) with vascular plant biodiversity and net carbon balance at the farm level. The method used to map HANPP related to livestock provisioning ecosystem services (ES) in the present study (lamb and wool), may be integrated into decision support systems. In this context, low HANPP values (<9%) promote sustainability-oriented economies within the region. Furthermore, keeping plant biodiversity and net carbon balance at the farm level could bring Patagonian export commodities recognition in international markets.

**Keywords:** rangeland; livestock; plant biodiversity; carbon balance; ecosystem services

## 1. Introduction

Human influence has played a large role in modifying natural ecosystems. Examples of modifications include global climate alteration, decline of wilderness areas, loss of biodiversity, and degradation of several ecosystem services (ES) [1–4]. The impact of humans on ecosystems and the definition of ES management strategies have gained recognition because these factors affect the supply of provisioning ES, the maintenance of ecosystem functions (regulation or support), and the conservation of biodiversity in anthropized environments [5,6]. Net primary production (NPP), an important metric of ecosystem functioning, represents the balance between gross biomass production from photosynthesis and plant respiration. It has been used as a proxy for the capacity of ecosystems to deliver

a range of other ES such as timber from native forests [7], livestock, firewood from silvopastoral systems [8], soil carbon [9] and nitrogen content [10], atmospheric regulation, water purification, and flow regulation [11].

Human appropriation of net primary productivity (HANPP) integrates ecological and socioeconomic perspectives on land use by quantifying the amount of NPP appropriated by society through biomass harvest from the ecosystem to the final consumption of biomass products [12–14]. HANPP is a measure of the impact of humans on biodiversity [12,15,16]. Global HANPP has been estimated for over 40 years using different definitions and increasingly sophisticated methods [17] with acceptable estimation errors, e.g., average 24% ± 10% (standard deviation) of potential NPP [14]. Rosas et al. [18] evaluated the potential biodiversity of vascular plant species across eight ecological areas in Southern Patagonia linked to environmental variables and ES supply. In addition, the magnitude of HANPP may affect the ecosystem carbon balance. For example, inappropriately implemented livestock grazing (overgrazing) systems can lead to a net release of $CO_2$ from depleting soil organic carbon stocks [9,19]. Peri [20] reported that carbon stock in grasslands decreased from 130 Mg C/ha under low grazing intensity to 50 Mg C/ha at sites with heavy stocking rates. The capacity of rangelands to produce biomass is one critical resource that sustains livestock production. Therefore, we hypothesize that HANPP may be in part determined by patterns of landscape plant diversity and carbon balance.

The main trade in the Santa Cruz province of Southern Patagonia is extensive sheep production, mostly Merino and Corriedale breeds reared for meat and wool. Production is based on natural grasslands where reproductive efficiency and animal performance is strongly dependent on environmental and management factors [21–24]. However, in Patagonia, there are more than 73.5 million ha with different degrees of desertification [25] due to a combination of extreme climate conditions and overgrazing in dry steppe areas. In these desertification areas, the soil loss rate ranged from 12.7 to 32.0 Mg/ha/year and soil carbon loss fluctuated from 85.3 to 250.1 kg C/ha/year [26]. Heavy and unsustainable grazing conditions threaten the future of livestock productivity. Therefore, regarding the long-term local economy, rangeland management should be based on maintaining biodiversity and regulating and supporting ES [6,27–29]. Previous research by Peri et al. [24] examined spatially explicit livestock provisioning ES assessments in Patagonia that were used to support decision-making. In this study, we evaluated the importance of HANPP related to livestock production that provides food, wool, income, and employment in areas such as Patagonia.

The main objective of the present study was to determine the spatial patterns of HANPP related to lamb and wool production from sheep farms across Santa Cruz to improve our understanding of interactions in human–environment systems at the regional scale. In addition, we examined the spatial relationship between HANPP and potential plant biodiversity and net carbon balance at the farm level under livestock land use across our study region.

## 2. Material and Methods

### 2.1. Characterization of the Study Area and Sheep Production

In the region, rainfall decreases from 800–1000 mm to 200 mm/year from west to east across the Andes Mountains, which acts as an orographic barrier to moist winds coming from the west. The mean annual precipitation to potential evapotranspiration ratio of the steppes fluctuates between 0.45 and 0.11, with marked soil water deficits in summer. Mean annual temperatures range from 5.5 to 8.0 °C. Winds, mainly from the west, consist of frequent gales reaching over 80 km/h in spring and summer. Local edaphic and topographic variations combined with a significant precipitation gradient substantially influence forage production on the grasslands.

The main trade in the study area is extensive sheep production, mostly with the Corriedale breed. Lamb production implies a particular nutritional requirement curve, with a higher demand before the start of winter to ensure pregnancy or May mating. There is also higher demand during winter until spring regrowth. The farm areas in this study

range from 20,000 to 35,000 ha with a breeding ewe flock size of 5000–22,500 head/farm. The vegetation of the steppe is dominated by grasses and sedges (*Bromus*, *Carex*, *Festuca gracillima*, *Hordeum*, *Jarava*, *Poa*, *Rytidosperma virescens*, *Trisetum*) with dwarf shrubs and herbs such as *Nardophyllum*, *Perezia*, *Azorella*, and *Nassauvia* admixed. The vegetation of the grass–shrub steppe is dominated by *Agrostis*, *Festuca*, *Hordeum*, and *Trisetum*. However, shrubs (*Adesmia*, *Chuquiraga*, *Junellia*, *Mulinum*, *Senecio*) are also frequent. The vegetation of shrubland or shrub–grass steppe sites is mainly dominated by tall shrubs such as *Berberis*, *Colliguaja intergerrima*, *Chuquiraga*, *Junellia*, *Lepidophyllum cupressiforme*, *Lycium*, and *Mulinum*, with grass-rich undergrowth including *Bromus*, *Hordeum jarava*, and *Poa*. The estimation of carrying capacity is based on the biomass production of short grasses and forbs that grow in the space among tussocks of each ecosystem. Requirements include 530 kg DM/year for 1 Corriedale ewe of 49 kg of live weight which represents a "Patagonian sheep unit equivalent (PSUE)" [30]. Overgrazing occurs when herbivore excess exceeds carrying capacity. The lambing rate (percent of ewes giving birth to a live lamb) fluctuates between 70 and 90%. The lamb growth rate fluctuates from birth to finishing after 100 days between 170 and 200 g/day [31].

For this study, we selected 72 permanent plots across Santa Cruz (Figure 1A) from the PEBANPA (Parcelas de Ecología y Biodiversidad de Ambientes Naturales en Patagonia Austral-Biodiversity and Ecological long-term plots in Southern Patagonia) network [32] to estimate HANPP at the regional level. The five ecosystem categories that contain the plots are: Mata Negra shrubland, Dry Magellanic steppe, Humid Magellanic Steppe, Central Plateau grasslands, and Andean grasslands (Figure 1B). Animal yield ranged from 0.25 to 0.69 g lamb/m$^2$/year (Figure 1C) and 0.10 to 0.19 g greasy wool/m$^2$/year (Figure 1D). The net primary production (NPP) data for the period of 2000–2015 with a resolution of 30 arcsec were acquired from the MOD17A3 data released by NASA's Earth Observation System Data and Information System [33]. In these ecosystems, NPP varied from 30.9 to 714.2 g C/m$^2$/year (Figure 1E). Net carbon balance at the farm level was estimated from the empirical farm data reported by Peri et al. [31], which used the anthropogenic greenhouse gas emissions and carbon footprint associated with sheep production in Southern Patagonia.

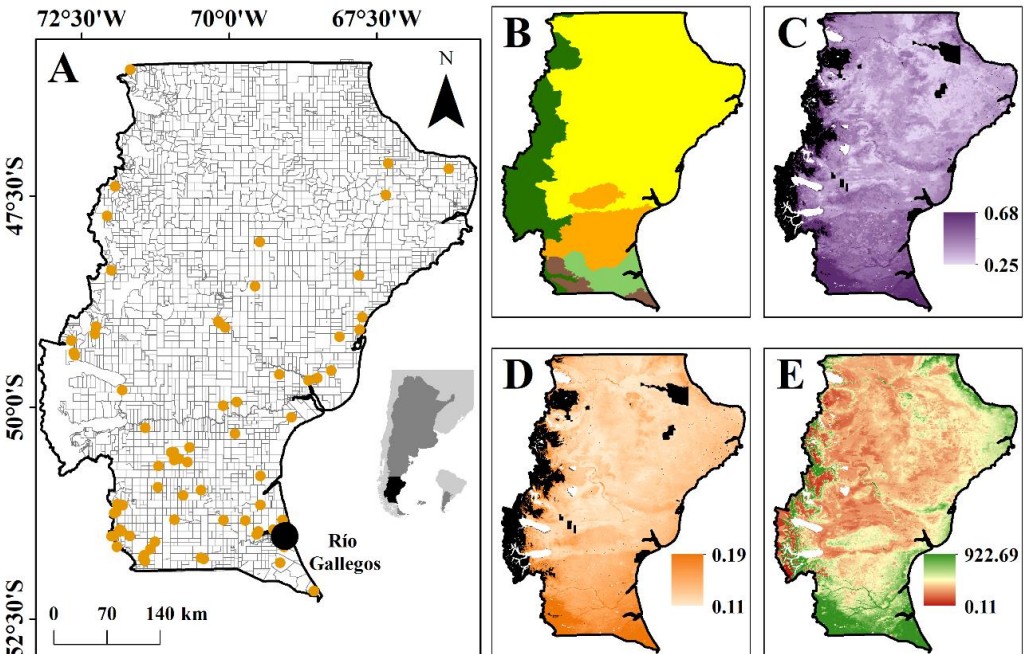

**Figure 1.** Study area. (**A**) Location of Argentina (dark grey), Santa Cruz (black), Rio Gallegos (black point), and sample sites (orange); (**B**) main ecological areas (dark green = Andean Region, brown = Humid Magellanic Steppe, green = Dry Magellanic Steppe, orange = Mata Negra Thicket,

and yellow = Central Plateau; (**C**) lamb yield (g lamb/m$^2$/year); high production = dark violet and low production = light violet (Peri et al. 2021); (**D**) wool yield (g greasy wool/m$^2$/year); high production = dark orange and low production = light orange (Peri et al. [24]); (**E**) net primary productivity (g C/m$^2$/year); high net primary productivity = green, and low net primary productivity = red [32]. Black areas represent NDVI < 0.05, elevation > 1200 m.a.s.l., *Nothofagus pumilio*, mixed evergreen forests, and natural protected networking areas where there are no livestock.

## 2.2. HANPP and the Relationship between Plant Biodiversity and Net Carbon Balance

We followed the HANPP concept as defined by Haberl et al. [13] and Krausmann et al. [34]. HANPP accounts for the NPP extracted by biomass harvest (HANPP*harv*) and the NPP losses due to land use change (HANPP*luc*). In our study area, HANPP*luc* is zero because there is no land conversion. HANPP*harv* is the quantity of carbon in biomass consumed by humans including used extraction. The used extraction in this work includes the forage consumed by livestock and the unused extraction includes the unused above and belowground grassland biomass. The HANPP was obtained by dividing the sum of biomass used in livestock products (lamb and wool) by NPP. The biomass used in livestock products was derived by multiplying the animal yield of each product (g lamb/m$^2$/year or g greasy wool/m$^2$/year) [24] by the carbon used during the production of a given product (g C/g lamb or g C/g greasy wool) derived from carbon footprint data [31]. The mean net primary production (NPP) (period 2000–2015) (g C/m$^2$/year) was obtained from the MODIS Net Primary Productivity MOD17A3H V6 product [33] (see Table A1 for more details about the data source). We consider both the above and belowground compartments of NPP and focus on the percent of the natural NPP appropriated by human activities within a location (hereafter HANPP).

We produced a final HANPP map for the entire province of Santa Cruz, where variables were integrated into the GIS using ArcMap 10.0 software. The map was adjusted to better represent the livestock activities. We applied a mask to remove areas with: (i) NDVI <0.05 that included glaciers, water bodies, rocks, and areas without vegetation cover [35]; (ii) ELE >1200 m.a.s.l. where sheep production was not conducted due to extreme climate; (iii) *Nothofagus pumilio* and mixed evergreen forests; and (iv) natural protected networking areas. The NDVI was downloaded from the MODIS collection [36]. Elevation was defined using a high-resolution digital elevation model from the shuttle radar topography mission [37]. Forests layers were obtained from SIT Santa Cruz (Sistema de Información Territorial, http://spm.sitsantacruz.gob.ar, accessed on 15 January 2022) and from protected area layers [38].

Finally, we analyzed the main ecological areas on the HANPP map to determine differences among the studied categories. We used hexagonal binning processes to divide the province into hexagons (*n* = 117) and for each hexagonal area we calculated the average values of HANPP. We tested the normality of the data considering standardized skewness and kurtosis using Stat-graphics software. HANPP data slightly deviated from a normal distribution when standardized skewness value was considered (−2.22). However, the standardized kurtosis value (−0.96) showed that data comes from a normal distribution [39]. A one-way ANOVA was used to analyze values considering different ecological areas (each hexagon = 250,000 ha). Fisher's test with post hoc mean comparisons using Tukey's test at *p* <0.05 was also calculated.

In addition, relationships between HANPP (dependent variable) and potential biodiversity of vascular plant and net carbon balance at the farm level (independent variables) were established by conducting simple linear regressions. We extracted the values of HANPP and potential biodiversity of vascular plants [18] using the evaluated 72 plots (Figure 1A). This map was created using the main plant species of each ecological area (Table A2). Scores varied from 0 to 100%, where low potential plant biodiversity was defined as values < 51%, medium was 52–62%, and high was >63% for the study area [18]. Net carbon balance at the farm level at each sampling location was estimated from Peri et al. [31]. Calculations considered several factors: emissions related to the use of fuel for internal

transport and electric generators; piped gas for cooking; coal and firewood for heating; fugitive emissions from household refrigerators and vehicle air conditioners; and the flows of GHGs into and out of animals, plants, and soils that occur on the farm.

## 3. Results

Main livestock production and site characteristics greatly changed through the main ecological areas (Table 1). The mean stocking rate varied significantly from 0.17 PSUE/ha in the Central Plateau ecological area to 0.80 PSUE/ha in the Andean Region (Table 1). Animal yield was higher in the Humid Magellanic Steppe (0.52 g lamb/m$^2$/year and 0.16 g greasy wool/m$^2$/year) compared with other ecological areas (Table 1), depending on climatic, topographic, and vegetation conditions from sheep farms across Santa Cruz [24]. Mean NPP showed the highest value (294.0 g C/m$^2$/year) in the Humid Magellanic Steppe and the lowest value (111.7 g C/m$^2$/year) occurred in the Central Plateau's ecological area (Table 1). Overgrazing reduced NPP by two thirds in most ecosystems. The estimated mean net carbon balance at the farm level fluctuated between −7.11 (Central Plateau) and 780.8 kg C/ha/year in the Andean Region (Table 1). While the negative C balance corresponded to sites with soil erosion loss (using a dendrogeomorphological method against datable exposed roots) greater than 10 Mg/ha/year [26], the highest positive net carbon balances occurred in farms located in more productive grasslands with *Nothofagus antarctica* forests in the Andean Region. Potential biodiversity of plant species greatly changed through the main ecological areas, where Mata Negra Thicket and Dry Magellanic Steppe ecological areas presented the highest values followed by the Humid Magellanic Steppe and the Central Plateau (Table 1).

**Table 1.** Mean and range (between brackets) values of livestock production and site characteristics in different ecological areas of Santa Cruz (Southern Patagonia, Argentina).

| Ecological Area | Stocking Rate (ewes/ha/year) | Net Primary Production (g C/m$^2$/year) | Lamb Yield (gr lamb/m$^2$/year) | Wool Yield (gr Greasy wool/m$^2$/year) | Net Carbon Balance (kg C/ha/year) | Potential Biodiversity of Plant Species (%) |
|---|---|---|---|---|---|---|
| Andean Region | 0.80 d (0.40–1.20) | 189.6 b (30.9–689.6) | 0.47 c (0.27–0.69) | 0.14 c (0.12–0.19) | 780.8 c (401.3–1073.1) | 45.73 a (34.3–76.1) |
| Humid Magellanic Steppe | 0.63 cd (0.25–0.78) | 294.0 c (78.2–714.2) | 0.52 d (0.35–0.65) | 0.16 d (0.12–0.18) | 203.7 b (130.8–244.2) | 58.59 bc (48.1–68.7) |
| Dry Magellanic Steppe | 0.44 bc (0.17–0.62) | 199.5 b (63.9–565.7) | 0.45 c (0.31–0.61) | 0.15 c (0.13–0.18) | 110.3 ab (18.8–228.1) | 68.83 c (58.2–75.3) |
| Mata Negra Thicket | 0.29 ab (0.14–0.52) | 142.4 a (57.5–432.3) | 0.40 b (0.28–0.55) | 0.13 b (0.12–0.16) | 168.5 b (42.6–330.5) | 69.06 c (54.1–77.1) |
| Central Plateau | 0.17 a (0.10–0.24) | 111.7 a (46.5–293.1) | 0.35 a (0.25-0.49) | 0.13 a (0.10–0.15) | −7.11 a (−93.6–103.9) | 58.60 b (46.1–76.4) |
| *p*-value | 47.13 (<0.001) | 34.29 (<0.0001) | 97.23 (<0.0001) | 90.45 (<0.0001) | 107.62 (<0.0001) | 21.39 (<0.0001) |

F(p), F-statistic and probability at *p* = 0.05. Values followed by different letters (a–d) in each column and for each variable are significantly different with Tukey's multiple range test at $p < 0.05$.

The results highlight the importance of HANPP related to livestock production that provides food and wool. Within the whole study region of Santa Cruz, livestock production accounted for an average of 11.35% of appropriated NPP, and HANPP ranged from 0.75 to 50% (Figure 2). The map of the HANPP model across Santa Cruz showed low values in the vegetation transition (ecotone) between *Nothofagus antarctica* forests and grasslands in the west, in the south, and in river valleys and wetlands where most productive rangelands dominate. High values were observed in the northwest and central areas of the province (Figure 2).

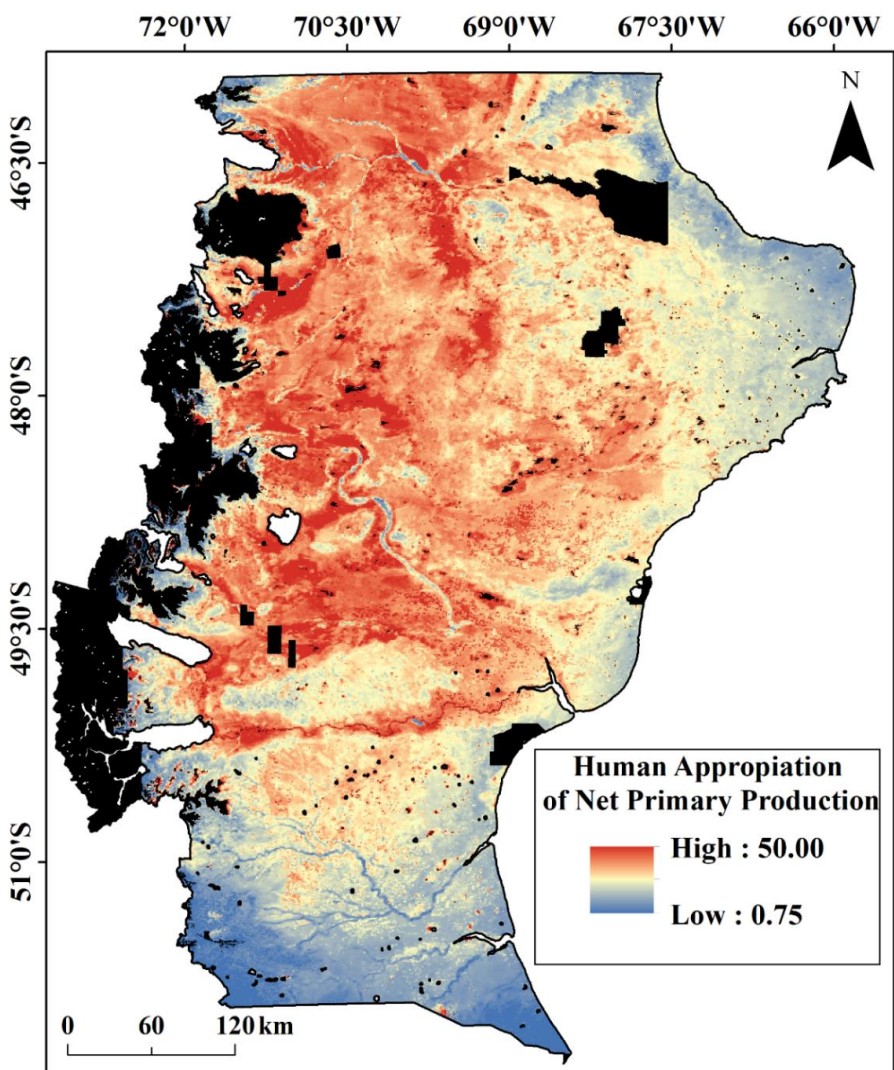

**Figure 2.** Human Appropriation of Net Primary Production (HANPP, %) in Santa Cruz, Argentina. Black areas represent NDVI < 0.05, elevation > 1200 m.a.s.l., and natural protected networking areas where there are no livestock.

There were differences in HANPP across vegetation types with mean values that varied from 3.93% in the grasslands on the Humid Magellanic Steppe to 12.33% in the Central Plateau (Table 2).

**Table 2.** Simple ANOVA analyses of human appropriation net primary production (HANPP) considering different ecological areas in Santa Cruz. *n* = number of hexagons extracted in the SIG for the different categories. Values followed by different letters (a–d) for each ecological area are significantly different with Tukey's multiple range test at $p < 0.05$.

| Ecological Areas | *n* | HANPP (%) |
|---|---|---|
| Andean Region | 17 | 8.73 bc |
| Humid Magellanic Steppe | 4 | 3.93 a |
| Dry Magellanic Steppe | 6 | 6.63 ab |
| Mata Negra Thicket | 13 | 9.92 c |
| Central Plateau | 77 | 12.33 d |
| F (*p*-value) | 117 | 25.85 (<0.001) |

HANPP was determined by patterns of landscape plant diversity and carbon balance. Simple linear regression analysis for HANPP evaluated in Southern Patagonia showed a negative linear relationship ($p < 0.05$) with potential vascular plant biodiversity (Figure 3A) and net carbon balance at the farm level (Figure 3B).

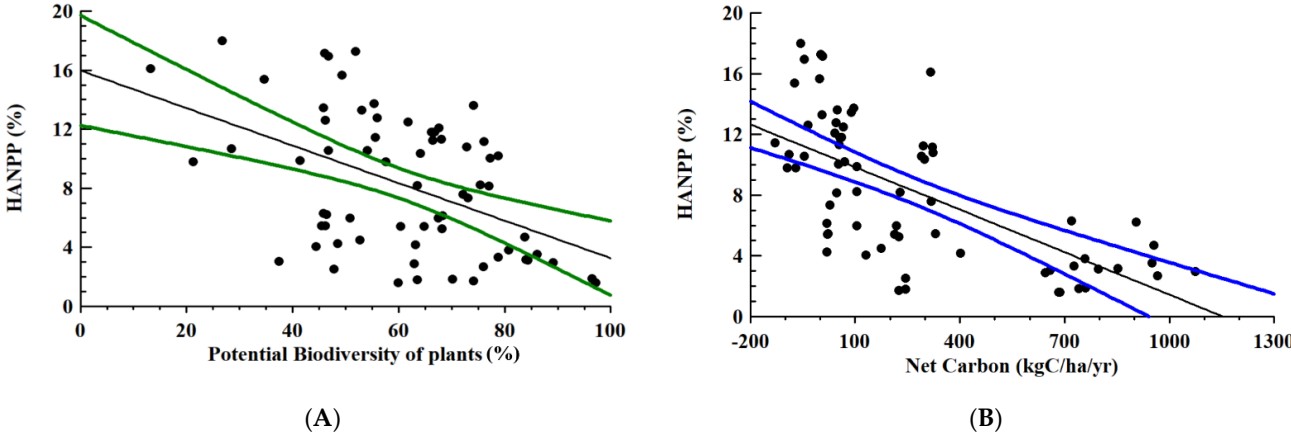

**(A)** **(B)**

**Figure 3.** Relationship between (**A**) human appropriation net primary production (HANPP) and potential biodiversity of vascular plant (HANPP = 15.98 − 0.1271 × potential biodiversity of vascular plants; $R^2$ = 0.22, ESE = 4.16). (**B**) HANPP and net carbon balance at the farm level (HANPP = 10.78 − 0.0093 × Net carbon; $R^2$ = 0.43, ESE = 3.56), Southern Patagonia, Argentina.

## 4. Discussion

In the present work, livestock production (lamb and wool) accounted for a mean regional value of 11.35% of appropriated NPP. We found that HANPP distribution across our study region aligned well with the range of global means HANPP estimated to be 15.6 Pg C/year, or 23.8% of NPP; 53% was attributed to the harvest of food and fiber; 40% to land-use-induced productivity changes, and 7% to human-induced fires [13]. This is consistent with previous research that highlighted several human-related activities in Southern Patagonia (e.g., livestock) negatively influenced the original ecosystems by modifying the plant biodiversity, soil properties, and structure [27,32,40]. The agricultural sector dominates global HANPP estimates, consisting of 84–86% of global NPP appropriated, 42–46% due to conversion to cropland, and 29–33% to grazing [34].

The map of HANPP across Santa Cruz showed low values in most grassland production sites—ecotone between *N. antarctica* forests and grasslands in the west, in the south at the Humid Magellanic Steppe, and in river valleys and wetlands—and high values in less productive areas (Central Plateau). Thus, production from grassland in good ecological condition had significantly lower HANPP but higher animal production (lamb and wool) values than overgrazed and ecologically degraded sites in the less productive grasslands. This is consistent with Lorel et al. [41] who reported for French agricultural landscapes higher levels of HANPP spatially congruent with low values of NPP. Conversely, high amounts of NPP were spatially matched with low values of HANPP. Variability in HANPP can be attributed to differences in grassland condition (forage quantity and quality) and lamb and wool production between farms as a result of long-term grazing management and climate conditions [24]. Heavy and unsustainable grazing conditions together with high HANPP values threaten the future of livestock productivity, therefore threatening the long-term wellbeing of the local economy [42].

HANPP depended on landscape plant diversity and net carbon balance. We found a negative linear relationship between HANPP and potential vascular plant biodiversity. Similarly, previous studies have reported an overall negative relationship between HANPP and biodiversity [12,14,16]. According to Franzluebbers [43], biomass being left on the field (low HANPP values) after harvest provides habitats to biodiversity conservation. Our results (low HANPP in productive grasslands) are consistent with the species-energy

hypothesis [44] that holds that energy availability in an ecosystem is positively related to species diversity. Thus, there is a positive relationship between ecological productivity and species richness [45]. In contrast, a reduction in energy availability in ecosystems by intense grazing (e.g., high HANPP values) is likely to affect the essential function which fosters species and habitat diversity in rangelands in Patagonia.

We determined a negative linear relationship between HANPP and net carbon balance at the farm level. Farms with higher productivity and low HANPP maximize their output from the resources invested and emissions linked to animals. The carbon content of biomass is closely associated with its energy content. Therefore, HANPP serves as an indicator of the human effects on flows of trophic energy in natural ecosystems and managed lands [13,46] since HANPP directly impacts biogeochemical cycles [15]. One factor that explains the negative relationship between HANPP and net carbon balance at farm level is the soil erosion rate in overgrazed grasslands. In the Pampean region (Argentina), Caride et al. [47] reported 15% loss of soil organic carbon by agriculture determining high levels of HANPP. These results implied that human activity generated net carbon losses in the entire region [48]. Similarly, in our work, while a low HANPP value of less than 3% determined net carbon balance at the farm level of around 1050 g C/m$^2$/year, high HANPP (more than 12%), negative carbon balance occurred in farms with a soil carbon loss rate from erosion greater than 50 kg C/ha/year [26]. Furthermore, HANPP often involves drastic changes in vegetation cover, whereas our region is mainly represented by the replacement of perennial-dominant grass species (e.g., *Festuca* sp., *Stipa* sp., *Poa* sp.) by bare ground or dwarf-shrubs due to overgrazing [49,50]. These structural changes reduce NPP at high HANPP values and therefore the carbon sequestration capacity of the ecosystem.

The method to map HANPP related to livestock provisioning ES in the present study (lamb and wool), may be integrated into decision support systems. The results of this study may help stakeholders and policy makers adopt sustainable management practices. This is especially the case in sites where HANPP are higher than 9% such as the Mata Negra Thicket and Central Plateau ecological areas. In these areas, a more uniform use of the rangelands at moderate stocking rates—together with supplementation strategies and subdivision of paddocks—would maintain or increase animal productivity, net carbon balance, and plant species richness and attenuate rangeland degradation [18,29,31,50]. Adjusting HANPP becomes relevant because rangelands in Patagonia not only support sheep farming (lamb and wool products), but also provide other benefits to society such as biodiversity conservation, regulating services (e.g., erosion and climate control), and cultural services (e.g., recreation, local identity, tourism) [6,7,9,28,51]. In this context, low HANPP values (<9%) promote sustainability-oriented economies within the region. Better plant biodiversity and net carbon balance at the farm level, could bring Patagonian export commodities recognition in international markets.

## 5. Conclusions

This study has provided HANPP values related to lamb and wool production in Southern Patagonian rangeland. The map of HANPP at regional scale (Santa Cruz province) provided an estimate of livestock use intensity showing that humans appropriated a mean value of 11.3% of the grasslands' NPP. Grasslands in good ecological condition had significantly lower HANPP but higher animal production (lamb and wool) values than overgrazed and ecologically degraded sites in the less productive grasslands. We found negative linear relationships between HANPP and vascular plant biodiversity and net carbon balance at the farm level. This can be attributed to differences in grassland conditions and animal production between farms because of long-term grazing management and climate conditions. The successful management of livestock becomes an important challenge to satisfying society's need for food and wool products under sustainable grassland management. We conclude that the HANPP framework provides useful indicators that should be integrated into future ecosystem service assessments. Future research is

needed to improve HANPP as a metric for understanding how resource extraction impacts conservation goals and grasslands ecosystem services.

**Author Contributions:** Conceptualization, P.L.P.; Data curation, Y.M.R.; Formal analysis, Y.M.R.; Funding acquisition, P.L.P.; Investigation, P.L.P. and G.M.P. All authors have read and agreed to the published version of the manuscript.

**Funding:** This research received no external funding.

**Institutional Review Board Statement:** Not applicable.

**Informed Consent Statement:** Not applicable.

**Data Availability Statement:** Not applicable.

**Conflicts of Interest:** The authors declare no conflict of interest.

## Appendix A

**Table A1.** Source data of livestock production characteristics and variables used for mapping and analysis of HANPP.

| Description | Unit | Data Source |
| --- | --- | --- |
| Stocking rate | ewes/ha/year | SIT Santa Cruz [1] |
| Net primary production | g C/m$^2$/year | MODIS [2] |
| Lamb yield | gr lamb/m$^2$/year | Peri et al. [24] |
| Wool yield | gr greasy wool/m$^2$/year | Peri et al. [24] |
| Carbon footprint of lamb production | kg $CO_2$-eq/kg lamb | Peri et al. [31] |
| Carbon footprint of wool production | kg $CO_2$-eq/kg wool | Peri et al. [31] |
| Net carbon balance | kg C/ha/year | Peri et al. [31] |
| Normalized difference vegetation index | dimensionless | MODIS [3] |
| Elevation | m.a.s.l. | DEM [4] |
| *Nothofagus pumilio* and mixed evergreen forests | occurrence | Forest map [1] |
| Natural protected networking | occurrence | Fasioli and Díaz [38] |
| Potential biodiversity of vascular plants | % | Rosas et al. [18] |

[1] SIT—Santa Cruz (http://www.sitsantacruz.gob.ar, accessed on 15 January 2022), [2] Running et al. [33], [3] ORNL DAAC [36], [4] Farr et al. [37].

**Table A2.** Taxonomy of the vascular plant species selected for the mapping of potential biodiversity in Santa Cruz province.

| Species | Code | Family |
| --- | --- | --- |
| *Acaena magellanica* | ACMA | Rosaceae |
| *Acaena poeppigiana* | ACPO | Rosaceae |
| *Adesmia volckmannii* | ADVO | Fabaceae |
| *Agrostis capillaris* | AGCA | Poaceae |
| *Agrostis perennans* | AGPE | Poaceae |
| *Anemone multifida* | ANMU | Ranunculaceae |
| *Armeria maritima* | ARMA | Plumbaginaceae |
| *Avenella flexuosa* | AVFL | Poaceae |
| *Azorella prolifera* | AZPR | Apiaceae |
| *Baccharis magellanica* | BAMA | Asteraceae |
| *Berberis empetrifolia* | BEEM | Berberidaceae |
| *Berberis microphylla* | BEMI | Berberidaceae |
| *Blechnum penna-marina* | BLPE | Blechnaceae |
| *Bromus setifolius* | BRSE | Poaceae |
| *Calceolaria uniflora* | CAUN | Calceolariaceae |
| *Carex andina* | CAAN | Cyperaceae |
| *Carex argentina* | CAAR | Cyperaceae |

**Table A2.** *Cont.*

| Species | Code | Family |
| --- | --- | --- |
| *Carex macloviana* | CAMA | Cyperaceae |
| *Chiliotrichum diffusum* | CHDI | Asteraceae |
| *Chuquiraga aurea* | CHAU | Asteraceae |
| *Chuquiraga avellanedae* | CHAV | Asteraceae |
| *Clinopodium darwinii* | CLDA | Lamiaceae |
| *Colobanthus subulatus* | COSU | Caryophyllaceae |
| *Empetrum rubrum* | EMRU | Ericaceae |
| *Ephedra chilensis* | EPCH | Ephedraceae |
| *Escallonia rubra* | ESRU | Escalloniaceae |
| *Festuca argentina* | FEAR | Poaceae |
| *Festuca gracillima* | FEGR | Poaceae |
| *Festuca magellanica* | FEMA | Poaceae |
| *Festuca pallescens* | FEPA | Poaceae |
| *Galium aparine* | GAAP | Rubiaceae |
| *Gaultheria mucronata* | GAMU | Ericaceae |
| *Hordeum comosum* | HOCO | Poaceae |
| *Hordeum pubiflorum* | HOPU | Poaceae |
| *Juncus balticus* | JUBA | Cyperaceae |
| *Lycium chilense* | LYCH | Solanaceae |
| *Microsteris gracilis* | MIGR | Polemoniaceae |
| *Mulguraea tridens* | MUTR | Verbenaceae |
| *Nardophyllum bryoides* | NABR | Asteraceae |
| *Nassauvia glomerulosa* | NAGL | Asteraceae |
| *Nassauvia ulicina* | NAUL | Asteraceae |
| *Osmorhiza chilensis* | OSCH | Apiaceae |
| *Pappostipa chrysophylla* | PACHR | Poaceae |
| *Pappostipa chubutensis* | PACH | Poaceae |
| *Pappostipa ibarii* | PAIB | Poaceae |
| *Pappostipa sorianoi* | PASO | Poaceae |
| *Perezia recurvata* | PERE | Asteraceae |
| *Poa lanuginosa* | POLA | Poaceae |
| *Poa ligularis* | POLI | Poaceae |
| *Poa spiciformis* | POSP | Poaceae |
| *Rytidosperma virescens* | RYVI | Poaceae |
| *Senecio filaginoides* | SEFI | Asteraceae |
| *Viola magellanica* | VIMA | Violaceae |

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
