# Peer review of "Human Appropriation of Net Primary Production Related to Livestock Provisioning Ecosystem Services in Southern Patagonia"

_sustainability, doi:10.3390/su14137617_

Round 1
Reviewer 1 Report
The manuscript aimed to determine the spatial patterns of human appropriation of net primary productivity related to lamb and wool production from sheep farms across Santa Cruz province. The subject is interesting and relevant in the context of sustainability, but important points should be improved before publication. The general and specific comments are described below.
General comments:
The results of biodiversity of plants, net carbon, livestock production (lamb and wool), net primary production and stocking rate are not analyzed and presented. It is difficult to understand the Discussion section without having the results.
Detailed information about the plant species representing the grassland biomass should be included in the manuscript.
The authors should revise the statistical analysis to properly analyze the data.
Specific comments:
Page 1, line 23. Please describe the meaning of the acronym ‘ES’.
Page 3, lines 87-90. Please include the description of the main plant species representing the grassland biomass.
Page 3, line 105. How did you define overgrazing?
Page 3, line 109. How was soil erosion loss obtained?
Page 4, lines 120-121. This phase is difficult to follow. Please, clarify. For which figure?
Page 4, line 143. It’s not clear the use of NDVI in the study. How was it obtained? Please clarify.
Page 5, line 150. Please correct: “one-way ANOVA was used to analyze”.
Page 5, lines 150-151. I’m not sure if the Fisher test with Tukey is the most appropriated to analyze data from this study. Data should be analyzed using non-parametric statistics. Please, justify the choice of the statistical tests used.
Page 6, line 167. What is ‘ecotone’?
Author Response
Reviewer #1
General comments:
The results of biodiversity of plants, net carbon, livestock production (lamb and wool), net primary production and stocking rate are not analyzed and presented. It is difficult to understand the Discussion section without having the results. Detailed information about the plant species representing the grassland biomass should be included in the manuscript. The authors should revise the statistical analysis to properly analyze the data.
We included two Appendices, one with source data of livestock production characteristics and variables used for mapping and analysis of HANPP (Appendix 1) and other with the taxonomy of the vascular plant species selected for the mapping of potential biodiversity in Santa Cruz province (Appendix 2). This determined four new references. The net carbon data and potential plant biodiversity are presented in Fig. 3. The statistical analysis has been revised.
Specific comments:
Page 1, line 23. Please describe the meaning of the acronym ‘ES’.
The meaning of the acronym ‘ES’ has been included
Page 3, lines 87-90. Please include the description of the main plant species representing the grassland biomass.
The description of main plant species representing the grassland biomass has been included. Also, we added one appendix related to the species list used in the potential biodiversity map.
Page 3, line 105. How did you define overgrazing?
Overgrazing occurs when herbivore excess over carrying capacity. The estimation of carrying capacity is based on the biomass production of short grasses and forbs that grow in the space among tussocks of each ecosystem and the requirements of 530 kg DM/yr for 1 Corriedale ewe of 49 kg of live weight. This overgrazing definition now has been included.
Page 3, line 109. How was soil erosion loss obtained?
The method to assess soil erosion rates has been included.
Page 4, lines 120-121. This phase is difficult to follow. Please, clarify. For which figure?
This has been completed
Page 4, line 143. It’s not clear the use of NDVI in the study. How was it obtained? Please clarify.
We clarified in the text and added cites related to NDVI, elevation and protected areas. The NDVI was downloaded from MODIS collection (ORNL DAAC, 2008), elevation was defined using a high-resolution digital elevation model from the shuttle radar topography mission (Farr et al., 2007), forests layer was obtained from SIT Santa Cruz (Sistema de Información Territorial, http://spm.sitsantacruz.gob.ar) and protected areas layer (Fasioli and Díaz, 2011).
Page 5, line 150. Please correct: “one-way ANOVA was used to analyze”.
This has been corrected.
Page 5, lines 150-151. I’m not sure if the Fisher test with Tukey is the most appropriated to analyze data from this study. Data should be analyzed using non-parametric statistics. Please, justify the choice of the statistical tests used.
We considered that Tuckey is appropriate in the data analysis due to their robustness. Also we added the skewness and kurtosis analysis in the text with a new cite.
Page 6, line 167. What is ‘ecotone’?
This has been clarified in the text.
Reviewer 2 Report
This paper aims to determine the spatial patterns of human appropriation of net primary production related to sheep farms in the province of Santa Cruz. It also examines the spatial relationship between HANPP and potential plant biodiversity and net carbon balance at the farm level.
Although the topic is potentially interesting, the article fails to bring new findings that would significantly improve knowledge on the topic. In addition, the data are not clearly presented, giving the impression that the article is almost a first draft of research. Finally, there is a lack of presentation of the source data, which should at least be included in the supplementary materials.
Comments:
L92 You have selected 72 permanent plots in Santa Cruz Province, but what kind of data were collected within these plots and where are the data from this sampling presented?
LL136-140 Have you compared the mean NPP obtained from MODIS with the actual NPP extracted from your permanent plots? Otherwise, I think I don't quite understand what the permanent plots are for.
L146 “(iii) natural protected networking areas”. This is the fourth object in the list, not the third.
LL149-151 You used ANOVA to compare HANPP values for ecological areas, but how were these values distributed? How can we say that these values are normally distributed if they are not presented anywhere in the research?
LL152-154 The same consideration as above applies to linear regression.
In addition, I found an inappropiate amount of self-citations: 18 out of 47 references.
Author Response
Reviewer #2
This paper aims to determine the spatial patterns of human appropriation of net primary production related to sheep farms in the province of Santa Cruz. It also examines the spatial relationship between HANPP and potential plant biodiversity and net carbon balance at the farm level. Although the topic is potentially interesting, the article fails to bring new findings that would significantly improve knowledge on the topic. In addition, the data are not clearly presented, giving the impression that the article is almost a first draft of research. Finally, there is a lack of presentation of the source data, which should at least be included in the supplementary materials.
We included an Appendix with source data of livestock production characteristics and variables used for mapping and analysis of HANPP (Appendix 1), and another with the taxonomy of the vascular plant species selected for the mapping of potential biodiversity in Santa Cruz province (Appendix 2).
Comments:
L92 You have selected 72 permanent plots in Santa Cruz Province, but what kind of data were collected within these plots and where are the data from this sampling presented?
The data from permanent plots were net carbon, livestock production (lamb and wool) and stocking rate. This information are in Table 1 and Fig. 3B.
LL136-140 Have you compared the mean NPP obtained from MODIS with the actual NPP extracted from your permanent plots? Otherwise, I think I don't quite understand what the permanent plots are for.
We did not conduct any ground-thrust validation for the MODIS Net Primary Productivity MOD17A3H V6 product, and we assumed that the output data of the model characterized the productivity of the studied landscape. Besides, we used a mask of NDVI to identify non-vegetated areas. These NDVI data were not used for HANPP calculation. These data were only used to remove some pixel from further analyses. The 72 permanent plots presented information about the net carbon estimated at the farm level. We used this data to evaluate their relationship with the HANPP in the simple linear regression analysis.
L146 “(iii) natural protected networking areas”. This is the fourth object in the list, not the third.
It has been corrected.
LL149-151 You used ANOVA to compare HANPP values for ecological areas, but how were these values distributed? How can we say that these values are normally distributed if they are not presented anywhere in the research? LL152-154 The same consideration as above applies to linear regression.
We added new information about the data normality, considering the skewness and kurtosis analysis in the text.
In addition, I found an inappropriate amount of self-citations: 18 out of 47 references.
This manuscript is a consequence of previous works. Following the advice, we deleted one self-citation.
Reviewer 3 Report
Comments to Editor:
The authors investigated the “Human Appropriation of Net Primary Production Related to Livestock Provisioning Ecosystem Services in Southern Patagonia.” The authors have not explained the research problem and research question clearly; therefore, it is strongly suggested to authors that they please add more relevant literature on their research question and then systematically link the research question with their main findings.
While I consider that the study and findings are potentially interesting, I have concerns that must be addressed in revising this manuscript. Based on my comments below, I recommend a minor revision of the manuscript.
Specific Comments
Abstract
The abstract needs to be revised (e. g., give details of the main results and link your results systematically and name the methods to achieve the goal/aim described in the background).
Introduction
It is strongly suggested that the authors please add more relevant literature on their research question.
Materials and Methods
This section is fine. I think no need for further improvement.
Results
Put an introductory/general statement about your main finding of each result as the first sentence.
Discussion
Authors should rethink what they write in the first paragraph and only summarize the main findings given the research questions.
Conclusion
It is strongly suggested to authors that please revise their conclusion and just give recommendations for future studies with shortcomings of your study.
Author Response
Reviewer #3
Abstract
The abstract needs to be revised (e. g., give details of the main results and link your results systematically and name the methods to achieve the goal/aim described in the background).
The abstract has been revised. The main results are indicated and we incorporated methods
Introduction
It is strongly suggested that the authors please add more relevant literature on their research question.
We have included a new reference (Jenkins, G.S.; Haberl, H.; Erb, K.; Nevai, A.L. Global human “predation” on plant growth and biomass. Glob. Ecol. Biogeogr., 2020, 29, 1052-1064).
Materials and Methods
This section is fine. I think no need for further improvement.
Results
Put an introductory/general statement about your main finding of each result as the first sentence.
A general statement of main finding of each result has been added.
Discussion
Authors should rethink what they write in the first paragraph and only summarize the main findings given the research questions.
The Discussion has been revised following the reviewer’s comment.
Conclusion
It is strongly suggested to authors that please revise their conclusion and just give recommendations for future studies with shortcomings of your study.
The Conclusion has been revised following the reviewer’s comment.
Round 2
Reviewer 1 Report
Dear authors,
The manuscript was improved, but the Results section still needs adjustments. The results of biodiversity of plants, net carbon, livestock production (lamb and wool), net primary production and stocking rate should be presented and analyzed among Ecological areas. It is difficult to follow the Discussion section without having these results.
Author Response
Now the results of biodiversity of plants, net carbon, livestock production (lamb and wool), net primary production and stocking rate are presented and analysed among Ecological areas. We incorporated a statistical analysis in Table 1. We move Table 1 to the Results section with new paragraphs to better follow the Discussion section.
Reviewer 2 Report
Dear authors,
the manuscript has been improved and can be considered for publication.
Author Response
We appreciate the reviewer’s comment
Round 3
Reviewer 1 Report
Dear authors,
The manuscript aimed to determine the spatial patterns of human appropriation of net primary productivity related to lamb and wool production from sheep farms across Santa Cruz province. The subject is interesting and relevant in the context of sustainability, and the manuscript was improved with inclusion of the results of biodiversity of plants, net carbon, livestock production (lamb and wool), net primary production and stocking rate. The manuscript can now be considered for publication in Sustainability.
This manuscript is a resubmission of an earlier submission. The following is a list of the peer review reports and author responses from that submission.